# Anxiety and Depression: What Do We Know of Neuropeptides?

**DOI:** 10.3390/bs12080262

**Published:** 2022-07-29

**Authors:** Ida Kupcova, Lubos Danisovic, Ivan Grgac, Stefan Harsanyi

**Affiliations:** 1Institute of Medical Biology, Genetics and Clinical Genetics, Faculty of Medicine, Comenius University in Bratislava, Sasinkova 4, 811 08 Bratislava, Slovakia; ida.kupcova@gmail.com (I.K.); lubos.danisovic@fmed.uniba.sk (L.D.); 2Institute of Anatomy, Faculty of Medicine, Comenius University in Bratislava, Sasinkova 4, 811 08 Bratislava, Slovakia; ivan.grgac@gmail.com

**Keywords:** anxiety, depression, neuropeptides, melanocortins, CRH, NPY, galanin family, spexin, Substance P

## Abstract

In modern society, there has been a rising trend of depression and anxiety. This trend heavily impacts the population’s mental health and thus contributes significantly to morbidity and, in the worst case, to suicides. Modern medicine, with many antidepressants and anxiolytics at hand, is still unable to achieve remission in many patients. The pathophysiology of depression and anxiety is still only marginally understood, which encouraged researchers to focus on neuropeptides, as they are a vast group of signaling molecules in the nervous system. Neuropeptides are involved in the regulation of many physiological functions. Some act as neuromodulators and are often co-released with neurotransmitters that allow for reciprocal communication between the brain and the body. Most studied in the past were the antidepressant and anxiolytic effects of oxytocin, vasopressin or neuropeptide Y and S, or Substance P. However, in recent years, more and more novel neuropeptides have been added to the list, with implications for the research and development of new targets, diagnostic elements, and even therapies to treat anxiety and depressive disorders. In this review, we take a close look at all currently studied neuropeptides, their related pathways, their roles in stress adaptation, and the etiology of anxiety and depression in humans and animal models. We will focus on the latest research and information regarding these associated neuropeptides and thus picture their potential uses in the future.

## 1. Introduction

Anxiety disorder and depressive disorders present a broad spectrum of diagnoses with a wide range of symptoms. The risk of depression is prevalent in younger patients (18–29 years old) but is not limited to this group, and even children with no indication of etiology suffer from the depressive disorder [1,2]. The Global Burden of Diseases, Injuries, and Risk Factors Study (GBD) 2019 showed that the two most disabling mental disorders were depressive and anxiety disorders [3]. Before 2020, the prevalence of the depressive disorder in European countries ranged from 2.58% to 10.33% with a mean overall prevalence of 6.38 [4]. Studies from the USA reported prevalences of 10.4% and 20.6% for 12-month and lifetime prevalences, respectively [5]. Global disease outbreaks in the past, or recently the COVID-19 pandemic, have significantly affected global mental health. However, psychological factors are often neglected, even though pandemics are psychological phenomena [6]. A systematic review on the effect of the COVID-19 pandemic on mental health in South Asian countries reported anxiety and depression rates of 34.1% to 41.3% in pooled patient samples [7]. In a recent meta-analysis on the global effect of the COVID-19 pandemic on the general population, Salari et al. reported a pooled prevalence of 31.9% for anxiety and 33.7% for depression, with a predominance in females [8]. Even before the pandemic, countries reported a rising incidence of depression in females [9]. The impact of the pandemic had an amplifying effect on mental health issues among the general population of low- and middle-income countries [10]. Adjusted for COVID-19, global prevalence showed similar results with elevation in anxiety (25.6%) and major depression cases (27.6%), which is a prominent reminder that mental health throughout the globe is an everlasting issue [11].

Both anxiety disorder and depressive disorders are of multifactorial origin composed of genetic, environmental, and psychosocial components. The exact molecular and neurobiological mechanisms are not yet thoroughly understood, but advances in molecular biology and genetics show promise in identifying the genetic component; meanwhile, neuroimaging and research on neuropeptides complement this knowledge. To date, some specific genotypes, the state and flexibility of the cardiovascular system, and function of the prefrontal and anterior cingulate cortexes have the ability to change the outcome of treatment [12]. Research on neurotransmitters mainly shows a contribution of the gamma-aminobutyric acid (GABA), serotonergic, noradrenergic, and dopaminergic systems [13]. There have been dozens of neurotransmitters and neuropeptides studied, but not all of them show a relationship with anxiety and depression. The list of 22 neuropeptides presented in Figure 1 was created from all the data in this article. Neuropeptides are separated into columns based on a strong or a moderate association with either anxiety or depression.

Neuropeptides are small peptide molecules with an approximate sequence length of 3 to 100 amino acids. Substance P was the first-ever neuropeptide to be identified by von Euler and Gaddum in 1931 and oxytocin the first ever to be sequenced. Since then, many more have been discovered and their effects are thoroughly studied. The location of secretion and area of effect are dependent on receptors associated with these molecules, which also determine their functions and associated pathways. Neuropeptides are not only a target for research or treatment but also viable diagnostic options for mental health disorders including, but not limited to, anxiety, depression, post-traumatic stress disorder (PTSD), or schizophrenia. These disorders have been a global health-related burden even before the COVID-19 pandemic, which only exacerbated and enhanced all contributing factors, causing mental health issues to become one of the dominant issues.

Several studies were aimed at addressing a multitude of individual objective and subjective symptoms, using diverse methods. These disorders exhibit, indeed, gender and occupational differences and higher strength in patients suffering from serious diseases [14,15,16]. Substances of abuse are often used to cope with anxiety and depression, leading to increased susceptibility to nicotinism, alcoholism, or drug misuse and addiction. Nicotine creates a sense of relaxation, and ethanol has sedative and anxiolytic effects, so there is no wonder these substances are often used in self-treatment. These relatively frequent habits are used to relieve self-perceived stress or anger, blunt feelings, and relieve the sensation of anxiety, even though over time they may have rather opposite effects. The use of these coping mechanisms is frequently seen in clinical practice even though a recent systematic review that collected 148 studies concerning the association of cigarette smoking behavior with depression and anxiety reported contrasting results because of inconsistencies in directions and strengths of the associations between these variables [17]. Many pathways with involvement in anxiety, depression, or addiction are mediated or affected by various neuropeptides, which could prove a viable path for research and the treatment of patients.

All the aforementioned facts urge more attention to mental health since these disorders have a rising tendency despite various available treatment methods. Research aimed not only at treatment but also at creating more effective diagnostic tools and markers are needed. Research suggests that neuropeptides could change the management of anxiety and depression [18]. The importance lies not only in the immediate future but also as a lesson from the past, to prepare for the future situation where population mobility reductions will be implemented and patients will be unable to visit a specialist or even benefit from telemedicine. This problem is of cardinal importance in all fields of clinical medicine.

The latest research in the field of neuropeptides presents promising results for the diagnosis, management, and treatment of patients with anxiety or depression. In this review, we take a close look at all currently studied neuropeptides, their related pathways, receptors, and their roles in the etiology of anxiety and depression in humans and animal models. We will focus on the latest research and information regarding these associated neuropeptides and thus picture their potential uses in the future.

## 2. Neuropeptides in Anxiety and Depression

Neuropeptides are small polypeptides of size ranging from 3 to 100 amino acids. Secretion is localized in neurons and the effect, depending on receptors, is local or distant. Affected are both neurons and other somatic cells, indicating a wide array of effects [19,20]. Some are synthesized as precursors and later cleaved into mature forms, while some have no post-translational changes [21]. Stress, pain, or emotional distress often leads to anxiety and depression. Studies of neuropeptides and their related receptors pose a high potential for a better understanding of the etiology/etiopathogenesis of these disorders, leading to better, more targeted surveillance and treatment, especially as diagnostic markers. In Table 1, we summarize the neuropeptides mentioned in this article with their related receptors and encoding genes. Data were acquired using the OMIM^®^ database accessed on 25 May 2022 [22].

Signaling axes (or pathways) studied in relation to anxiety and depression are the hypothalamic–pituitary–adrenal (HPA) axis, the hypothalamic–pituitary–gonadal (HPG) axis, and the gut–brain axis (GBA). The GBA is newer, and less thoroughly studied, although reports indicate that the microbiome and gut hormones may play a role in the etiology of depression [23]. More studied are the HPA and HPG axes. Interactions of these axes are presented in Figure 2.

The HPA axis affects many different systems in the mammalian body, with a modern link to neuropeptides, psychiatry, and neurology, on which we will focus in this review [24]. The HPA axis is controlled by the limbic system depending on stimuli (stressors) received by the latter, and so inducing release of the CRH. CRH stimulates the release of ACTH (adrenocorticotropic hormone), which affects the adrenal glands and the production of glucocorticoids. Stress-adaptation responses are mediated by corticosterone in rodents and by cortisol in humans.

Circulating levels of gluco- and mineralo-corticoids are essential for negative feedback on the hypothalamus, which has been reported faulty in patients suffering from depression [25]. A systematic review and meta-analysis on the effects of aging and the HPA axis in patients with depression reported that older patients suffering from depression showed a higher degree of dysregulation in the HPA axis [26]. A bipolar disorder- (BD) focused systematic review and meta-analysis found a possible association of progressive HPA axis dysfunction with the cognitive deterioration of BD patients, which seemed rather a risk factor than a determinant, but nonetheless an important one [27]. In a recent review, an association between early-life stress in the HPA axis and anxiety has been found, followed by hyperactivity in the axis acting as a risk factor for relapses [28]. Effects of the microbiota on the HPA axis, stress response, anxiety-like behavior, endocrine abnormalities, and wider neuropsychiatric disorders have been observed, calling for further investigation [29,30,31,32]. In a systematic review, Juruena et al., based on the activity of the HPA axis, found a difference between melancholic and atypical depressive subtypes [33]. However, they attributed these findings to hypercortisolism in melancholia and rather normal, than decreased, function in atypical depression [33].

The HPG axis is represented by neurons of the gonadotropin-releasing hormone (GnRH), which in mammals is a 10-amino acid peptide. GnRH stimulates the synthesis and secretion of luteinizing hormone (LH) and follicle-stimulating hormone (FSH), thus having an important role in the regulation of fertility and reproduction. GnRH is inhibited by the gonadotropin-inhibitory hormone (GnIH), which belongs to the RFRP family. The RFRPs (RF-amide-related peptides) act in regulatory regard to the HPG axis [34]. Studies on the GnIH have been associated with different behaviors in male rats [35]. Stress-induced increase in adrenal glucocorticoids increased the RFRPs, which contributed to the suppression of reproductive function through the HPG axis [36]. In birds, a small dose of interfering RNA targeted against the GnIH precursor mRNA caused elevation in aggressive and sexual behaviors [37]. Iwasa et al. reviewed the field of stress-induced reproductive dysfunction and found an association with the GnIH/RFRP-3 system [38]. These findings are important as stress, whether physical or emotional, is related to anxiety and depression in a reciprocal character, and one may affect or over time induce the other. The use of GnRH analogs caused upregulation in the expression of phoenixin in hypothalamic, hypophyseal, and ovarian regions, while the GPR173 expressions were downregulated in the hypothalamus and pituitary [39].

### 2.1. Oxytocin

Oxytocin (OXT) is composed of nine amino acids and was the first polypeptide hormone to be sequenced. OXT is synthesized in the supraoptic and paraventricular nuclei of the hypothalamus [40]. It became first known for its role in lactation, parturition, and maternal behavior, yet over the last several decades it has also been implicated in memory, regulation of anxiety, mood, and social behavior [41]. In reaction to acute stress, oxytocin is released from the nucleus centralis of the amygdala, which leads to the local activation of GABA-ergic interneurons [42]. While the amygdala is responsible for acute stress reactions, stria terminalis is the structure involved in the transition to chronic anxious states. A portion of oxytocinergic neurons from the periventricular nucleus projects directly into this structure [43]. The anterior cingulate cortex likely facilitates learning and social adaptation [44,45]. Oxytocin receptors (OXTR) are present across the whole brain, and it seems that the research of polymorphisms and epigenetic markers for OXTR is a promising area for future research [46]. Myers et al. reported two single nucleotide polymorphisms (SNP) in the OXTR gene in depressed patients [41]. Costa et al. described a relationship between separation anxiety in adulthood and SNP rs53576 of the third intron of the OXTR gene, which seems of substance for the epigenetic regulation of this gene [47]. An allele of rs53576 is also related to increased suicide risk, according to some authors [48]. Oxytocin interacts with various neurotransmitters and neuroendocrine systems [49]. The most prominent ones are the serotonergic and GABA-ergic systems and the HPA axis. Mottolese et al. have proven the interplay between oxytocin and serotonin using a 5-HT1A receptor antagonist 2’methoxyphenyl-fluorobenzamidoethylpiperazine (MPFF). Intranasal administration of oxytocin resulted in increased levels of MPPF in nucleus dorsalis raphes, the right amygdala-hippocampus-parahippocampus complex, insula, and right medioventral prefrontal cortex, measured using PET MRI [50]. The interplay between the oxytocinergic and serotoninergic systems is further supported by the fact that MDMA (3,4-methylenedioxymethamphetamine) causes oxytocin-mediated behavioral changes via serotoninergic 5HT1A receptors [51]. Furthermore, oxytocin is also connected to the HPA axis. Part of the neurons in the paraventricular nucleus co-expresses oxytocin and CRH, a portion of CRH-releasing neurons expresses mRNA for the OXTR, and clusters of OXTR express CRH receptor CRHR2. Centrally administered oxytocin reduces the excitability of CRH neurons, inhibits their spontaneous excitability, and lowers the expression of CRH mRNA and neuroendocrine response to stress [42]. In animals, the central administration of oxytocin reduces anxiety-like behaviors and lowers plasmatic cortisol levels [52]. Furthermore, the medial prefrontal cortex contains oxytocinergic interneurons synthetizing CRH binding protein (CRHBP) and GABA. In male rodents, the stimulation of these interneurons leads to the release of CRHBP and the reduction of CRH activity and anxiety, while in female rodents CRHBP was not able to effectively inhibit CRH activity, which may be due to higher levels of CRH in paraventricular nuclei (PVN) [53]. Antagonistic activity on GABAA receptors in the PVN blocks the suppression of stress-induced CRH release and the application of oxytocin into the PVN results in a significant increase in GABA release from the PVN [52]. Meisenberg was the first to describe the antidepressant-like effect of oxytocin when oxytocin application had an effect similar to the tricyclic antidepressant imipramine in rodents—it resulted in reduced immobility during the forced swim test [51]. Later, Scantamburlo et al. reported an inverse relationship between plasma oxytocin levels and the severity of symptoms of depression and anxiety in depressed patients [54]. Jobst et al. measured plasmatic levels of oxytocin in chronically depressed patients before and after psychotherapy. In their research, increased levels of oxytocin correlated with a reduction of subjectively reported depressive symptoms according to the Beck Depression Inventory-II [55]. Lancaster et al. found that higher endogenous oxytocin levels are associated with a reduced central amygdala volume and blood oxygen level-dependent activity in response to aversive stimuli [56]. However, the relationship between oxytocin and depressive and anxiety symptoms is not as straightforward as it may seem, and the literature is full of contradictory results. For example, Parker et al. found increased oxytocin levels in depressed subjects, and more recently Tabak et al., examining the role of oxytocin in social anxiety in humans, reported increased plasma concentrations of oxytocin following the Trier Social Stress Test (TSST), but this increase was specific to women, and that social anxiety moderated the OXT increase with more socially anxious participants showing greater increases than those with lower levels of social anxiety [57]. This can be the result of the so-called oxytocin paradox, which has been described as the context-dependent feedback loop of OXT signaling based on socio-psychological settings leading to positive or adverse psychological consequences [58]. The functioning of oxytocin varies according to contextual factors and individual differences, including gender, age, and psychosocial functioning [59]. In the past years, research allowed us to better understand signaling ways affected by OXT and its complex regulation, and further multi-peptide-centered research will reveal this intricate web of regulation.

### 2.2. Vasopressin

Vasopressin, also known as the antidiuretic hormone (ADH) or arginine vasopressin (AVP), consists of 9 amino acids, belongs to nonapeptides, and is synthesized by magnocellular cells of the hypothalamic supraoptic nucleus and PVN whose axons project to the posterior pituitary [60,61]. After release into the bloodstream, it is transported predominantly to kidneys and blood vessels, where it regulates physiological functions (e.g., resorption of water, vasoconstriction, …). Vasopressin has amino acid sequences and tertiary structures similar to oxytocin, probably due to the shared origin of their encoding genes [62]. In addition, it is possible that an interplay between oxytocin and vasopressin is involved in mood and stress-response regulation [63]. It binds to four types of G-protein-coupled receptors: AVPR1A, AVPR1B, AVPR2 (for clarity reasons further as V1A, V1B, and V2, respectively), and oxytocin receptor [64]. V2 receptor binds to vasopressin in the kidney, where it participates in water homeostasis [65]. On the other hand, V1B and V1A receptor subtypes are promising sites of interest for psychiatry. The V1B subtype is highly expressed in the anterior pituitary gland, where it stimulates corticotropin release and is also localized in the amygdala, hippocampus, and hypothalamus [66]. Purba et al. already in 1996 reported increased numbers of V1B receptors in the postmortem analysis of brains of depressed patients compared to controls [67]. In depression and anxiety, vasopressin binds to V1 receptors in the brain. Hoghson et al. used V1B-30N-A potent V1B receptor antagonist in rat pups to examine the potential role of V1B in anxiety. They reported that V1B-30N had a dose-dependent anxiolytic effect in the separation-induced vocalization test in rat pups, moreover without causing sedation [64]. The interaction between vasopressin and oxytocin has already been mentioned, but vasopressin co-operates at least with one other neuromodulator system—the serotonergic. Ishizuka et al. used V1B knockout mice treated with a selective serotonin reuptake inhibitor (SSRI) and serotonin noradrenaline reuptake inhibitor (SNRI) to examine whether at least part of the SSRI´s effects might be mediated via the V1B receptor. The effects were evaluated in experiments using an elevated plus-maze (EPM) test and a hole-board (HB) test. Chronic treatment of V1bR KO mice with SSRI did not change the amount of time spent on the open arms, the number of head dips, or the number of readings, while chronic treatment with SNRI significantly increased the time spent on the open arms and the number of head dips. These results suggest that the anxiety action of 5-HT reuptake inhibitors might partly involve V1bR regulating the anxiety behaviors [66]. While the V1B receptor likely plays a role in depression and anxiety, the V1A subtype seems to be related to a different type of psychopathology. Vollegbregt et al. in their study reported an association between RS3 microsatellite repeats within the promoter region of the V1A gene and extreme childhood aggression, while no association was found between childhood aggression and RS1 repeats [68]. However, in addition to factors related to the V1 receptor, the levels of vasopressin also likely play a role in the pathophysiology of depression and anxiety. It was shown that chronically elevated plasma vasopressin levels may induce depressive symptomatology [69]. Another research project pointing to the positive correlation between vasopressin levels and depression was performed by Mlynarik et al., which showed reduced depression-like behavior in vasopressin-deficient Battleboro rats during the forced swim test [70]. The results of human studies agree with animal research. Goekoop et al. measured plasma levels of vasopressin in 89 patients with a highly anxious-retarded subtype of depression with a family history of depression. Depression with above-normal plasma AVP, as well as familial depression with above-normal plasma AVP, showed a high correlation between anxiety and retardation and this correlation was significantly higher than that found in the depressed patient control groups. The data support the delimitation of a largely familial depression with above-normal plasma AVP, vasopressinergic activation of the HPA axis, and a variable anxious-retarded phenotype [71]. As already mentioned above, the current situation regarding the efficient treatment of anxiety and depression is far from satisfactory despite the fact that there are multiple pharmacological agents on the market. The rate at which novel antidepressants are developed and introduced into clinical practice is very slow. This motivates research into different treatment modalities, of which V1B antagonists seem to be one promising option [72,73].

### 2.3. Melanocortins

Melanocortins are a group of peptide hormones derived from pro-opiomelanocortin (POMC). This group includes the adrenocorticotropic hormone (ACTH), alfa-, beta-, and gamma-melanocyte stimulating hormones (MSH), beta-endorphin, and corticotropin-like intermediate peptide (CLIP), the adrenocorticotropic hormone fragment [74]. Melanocortin receptors are associated with obesity, erectile dysfunction, cachexia, pain, depression, and anxiety [75]. Melanocortins work through the melanocortin receptors. The five melanocortin receptors are GPCRs identified as MC1 to five receptors. To date, only MC3R and MC4R with their ligand, the alfa-MSH (also gamma-MSH in the case of MC3R) were connected to anxiety and depression [76]. The main expression sites of melanocortin neurons are located in the brainstem and the hypothalamus, in the arcuate nucleus (ARC), closely communicating with AgRP- and Neuropeptide Y- (NPY) expressing neurons, thus indicating their participation in the system [77,78]. The alpha-MSH was reported to suppress the effects of NPY, thus reducing its antidepressant effect [79]. The administration of alpha-MSH antagonist with NPY shows a synergistic anxiolytic effect [80]. The inhibition of MC4 receptors in the dorsal raphe nucleus (DRN) using alpha-MSH induces anxiety and depression and reduces feeding in mice [81]. PVN, ARC, and DRN are predominant locations of the alpha-MSH effect [82]. The circle of regulation in the melanocortin system is closed by the receptor antagonist, the Agouti-related peptide (AgRP), with an orexigenic effect [83]. Chronic administration of a high-fat diet blunts AgRP response to anxiety and depression signals, as well as hunger by reducing GABA-ergic outputs from AgRP aimed at MC4R. However, GABA-ergic stimulation and suppression of the 5-HT3R within the MC4R neurons in the bed nucleus causes cessation of the effect of the high-fat diet-induced anxiety and depression [84]. The combined effect also reduces food intake and thus body weight. These findings indicate that initially AgRP regulates appetite, but later with a loss of effect, MC4Rs are a viable option for further research on anxiety and depression treatment [85,86]. Further associations are discussed in individual sections. A specific regulation protein, which affects the melanocortin system, is the melanocyte-stimulating hormone release-inhibiting factor-1 (MIF-1) with its analog Nemifitide. Both show low affinity to μ-opioid receptors and function as blockers of alpha-MSH release, thus reducing its inhibitory effect on NPY. Preclinical studies have shown effects in depressive disorder, but further research into this particular peptide is necessary to prove its efficacy.

### 2.4. Corticotropin-Releasing Hormone

Corticotropin-releasing hormone (CRH) is also known as the corticotropin-releasing factor (CRF); however, for precise nomenclature reasons, further on, the term CRH will be used. CRH is the central regulator of the hypothalamic–pituitary–adrenal (HPA) axis, which is the main organizer of the body’s response to stress. CRH consists of 41 amino acids and binds to CRHR1 and CRHR2, two GPCRs (not counting alpha and beta splicing variants), both in the central nervous system and the R2 also in peripheral tissues [87]. CHR receptors are expressed in the pituitary, amygdala, hippocampus, brain stem, and cingulate cortex (20). The CRH system consists of CRH itself, then Urocortin 1, 2, and 3, which all serve as ligands for the CRH receptors [88]. Effects of this system lie not only in the mediation of stress-related responses but also in the regulation of inflammation, with a close relationship with inflammation-related pain, as macrophages, monocytes, or mast cells do express both CRH receptors, thus serving as targets of this system [89]. CRH-induced stress-related responses through the HPA axis have a wide area of effect, recently measured in human hair follicles connected to hair loss interestingly antagonizable by caffeine [90]. Here we can observe how deteriorative an effect the CRH-HPA system can have on the human body and how broad the effect is. It is not surprising that many human and animal studies report CRHs’ effect on the HPA axis regulation. Chronic stress activates the CRH-related signaling in the bed nucleus and induces maladaptive behavior in mice [91]. This goes in line with observed HPA axis dysfunction in patients with major depression or schizophrenia, where the continuous effect of CRH on the HPA axis causes disbalance (increased cortisol, increased ACTH, reduced feedback) and results in a pathology [92,93]. This disbalance is targeted by antidepressants, which after a period of time have the potential to revert changes in HPA axis hyperactivation. The CRH system is long associated with serotonergic mediation, emotional disbalance, behavioral changes, anxiety, and depression [94,95,96]. CRHR1 gene variants have been linked to a higher susceptibility to depression or panic disorder [97,98]. A sex-specific stress reaction in females has been observed in an animal model, with sex hormones affecting CRH regulation [99]. Transgenic mice with induced CRH expression show hyperactivation of the HPA axis and an increase in stress-mediated processes and behaviors, treatable with CRHR1 antagonists [100].

Further members of the CRH family are the urocortins. Urocortin 1, 2, and 3 are a group of three peptides (UCN1, UCN2, and UCN3). UCN1 is a 40-amino acid peptide, while UCN2 and 3 are paralogous 38-amino acid peptides that bind to CRHR2 [101]. UCN1 is produced by the hypothalamus, substantia nigra, and the pituitary gland and has a high affinity for both CRH receptors, suggesting its role in behavior and interestingly also regulation of food intake [102]. The intracerebrovascular injection of UCN1 has anxiogenic and “anti-social” effects antagonizable by CRHR1 antagonists [103,104]. However, the CRHR1 antagonist had no effects on social interaction on its own but antagonized the decreases in social interaction induced by stress or UCN1 [105]. Summarized by Tanaka et al., UCN2 and UCN3 show an association with depression-like behavior, where stimulation by UCN1 (CRHR1-2) is antagonized by UCN2 and UCN3 (CRHR2), thus resulting in no effect [106].

The latest members associated with the CRH family are four teneurins, or teneurin C-terminal associated peptides (TACPs), which share homology with the amino acid sequence of CRH and bind to latrophilins (a group of highly conserved GPCRs). Their effect is tissue-specific, and reports show an ability of TACP-1 to reduce anxiety, addiction, and depression in stress-induced behaviors [107].

On the other hand, studies and preclinical models also show mixed results [108]. In some cases, CRH or TACP receptor antagonists have different effects on anxiety, based on the animal test line, the baseline of depression, or perceived anxiety or the treatment regimen, but also depending on the test, as these therapeutics do not work under basal conditions. However, if depression- or anxiety-related symptoms or signaling are present, these therapeutics do have a significant effect [109]. In other words, a stressor (depression or anxiety) must be present for antidepressants and anxiolytics to work [110]. These conclusions are not surprising, as urocortins and their over-expression in CRH-deficient individuals are often induced as a compensatory mechanism in absence of CRH. The same result is for models with CRH over-expression antagonizable by CRHR1 antagonists, where some lines do exhibit higher anxiety and vulnerability to stress, but in other lines that do not exhibit these traits, a UCN1 downregulation was found [111,112].

### 2.5. Pituitary Adenylate Cyclase-Activating Peptide

Pituitary adenylate cyclase-activating peptide (PACAP) is a neuropeptide existing in two isoforms, either 27 or 38 amino acids long. PACAP is relatively well-conserved across many species [113]. Wide expression in central and peripheral tissues indicates a role in multiple physiological functions, such as the modulation of nociception, regulation of prolactin release, and food intake, along with stress, anxiety, and depression [114,115]. PACAP has the highest affinity for the ADCYAP1R1 receptor, known also as the PAC1 receptor [116]. Studies have associated this neuropeptide with stress-related mood disorders and adaptation [19,117]. PASAP is predominately expressed in the amygdala and the bed nucleus [118,119]. Dore et al. observed pro-depressive and anxiety-like effects on rats after intracerebroventricular PACAP administration [120]. Human studies show an association of PACAP/PAC1 receptor with behavioral and endocrine responses, also in neuropsychiatric disorders such as schizophrenia or PTSD [121,122,123]. A recent animal model of PACAP-mutated mice under chronic variable mild stress exhibited viable results for future depression-targeted tests [124]. PACAP-KO mice exhibit decreased c-Fos expression, which shows a close connection not only with CRH and wider melanocortin signaling [125,126]. These results go in line with previous results on PACAP’s influence on CRH [127]. Additionally, a study on the consequences of chronic stress on PACAP expression in the bed nucleus goes in hand with CRH, indicating that synergic modulation exists in anxiety-like behavior [128]. The anxiety-associated effect of PACAP was further studied in mice, where the induction of PACAP-firing neurons aimed at the bed nucleus enhanced anxiety-like behavior [129].

### 2.6. Melanin-Concentrating Hormone

Melanin-concentrating hormone (MCH), or pro-melanin-concentrating hormone (PMCH), exists in a cyclic form and consists of 19 amino acids. MCH binds to two GPCRs, the MCH-1R and MCH-2R, of which only the 1R type is expressed in rodents, while both are present in humans [130]. Receptors are widely expressed in the brain, but MCH secretion, depending on species, is limited to sections of the hypothalamus and zona increta [131]. MCH is involved in the regulation of feeding and energy homeostasis, also stress-related adaptation, emotions, and cognitive functions, some of which are due to the MCH interaction with neurotransmitters of serotonergic and cholinergic systems [132,133]. MCH reverses activation of the HPA axis and also interacts with associated serotonergic and cholinergic pathways [134,135,136]. MCH-1R exhibits a potential for anxiety and depression treatment because even in animal models, MCH-1R knockouts exhibit an anxiety-resistant phenotype [137,138]. This potential has been studied by the administration of different MCH-1R antagonists that produced antidepressant results in rats [139,140]. MCH-1R receptor antagonists infused in the nucleus accumbens are producing faster anxiolytic and antidepressant effects in mice and rats compared to the selective serotonin reuptake inhibitors (SSRIs), while receptor agonists produce opposite effects [141]. This happens concurrently with induced sensitivity of dopamine D2 and D3 receptors [141]. The nucleus accumbens has a relation to the mediation of relative motivation for rewards. A recent animal study observed reduced stress-induced anxiety and depression after intranasal administration of MCH [142]. These results in animal studies are further proof of the important role of MCH in the etiology of anxiety and depression with connections to the HPA axis via CRH, where the MCH-1R antagonists act as inhibitors of this connection.

### 2.7. Beta-Endorphin

Beta-endorphin (β-endorphin) is a 31-amino acid endogenous opioid neuropeptide that binds to opioid receptors. The mature form is derived from precursor POMC. There have been conflicting findings in research regarding its potential role in depression, but recent studies show decreased β-endorphin activity during negative moods, and a decreased quantity of its μ receptors has been found in the brains of depressed suicide victims [143]. There is also a suggestion that its plasmatic levels might correlate with the response to depression treatment [144]. There has also been evidence that the opioid system is involved in the regulation of anxiety, and its activation has been demonstrated to have an anxiolytic effect in humans [145]. Savic et al. created a model on plasma beta-endorphin in which only anxiety and hyperarousal were directly associated with peripheral beta-endorphin fluctuations [146]. Patients with Meniere’s disease or dyscirculatory encephalopathy with the vestibular ataxic syndrome were examined for beta-endorphin levels before and after pharmacological and physical rehabilitative treatment where the levels correlated with the degree of anxiety and depression, but not the vestibular-related symptoms [147]. The intracerebroventricular administration of beta-endorphin antagonized suppressive effects of alfa-MSH on food intake and weight gain for a limited time and later lost effect, which indicates an intricate cross-regulation in the melanocortin system [148].

In a recent study on the effects of auricular point sticking therapy in patients undergoing partial lung resection, beta-endorphin concentration was used as a marker of peri- and post-operative pain, anxiety, and depression. Authors attributed the analgesic mechanism to the increase in plasma concentration of β-endorphin [149]. In a model on opiate-mediated analgesia in rats suffering from osteoarthritis (OA) -like symptoms combined with normal or anxiety-like behavior, baseline plasma levels of β-endorphin were significantly lower in the OA + anxiety group [150]. However, this effect was also observed after an intra-articular injection of saline in both groups, which is possible since, in a meta-analysis, the efficacy of saline injections in articular pain management of OA patients was found to be on par with other injectable options [151].

### 2.8. Neuropeptide Y

Neuropeptide Y (NPY), a 36-amino acid neuropeptide, has a wide distribution in the central nervous system (CNS) with connections to the melanocortin system. NPY belongs to the most conserved proteins in evolution. In the past, NPY levels were linked to affective disorders, and measured plasma levels were low in suicidal depressed patients [152,153]. The effect of NPY through its receptors has been reviewed by Morales-Medina et al., with the conclusion that animal models on Y1 and Y2 receptors provided robust data on their role in emotional responses and stress [154]. Holzer et al. reported the role of NPY, peptide YY (PYY), and pancreatic polypeptide (PP) on depression-related behavior through the gut–brain axis [155]. A recent study on a murine model demonstrated that depression increased IL6 levels and promoted myeloid cell infiltration by a sympathetic-NPY signal [156]. A different murine model reported depression-like behavior associated with the decreased expression of NPY in the hypothalamus and hippocampus, while also inducing changes in the microbiome and brain metabolome in mice on a high-fat diet [157]. NPY is partially connected to the HPG axis via kisspeptin neuron fibers on the ARC neurons that express POMC, the precursor of melanocortins, and indirectly inhibits the ARC neurons expressing NPY. A review by Dr. Domin suggested neuroprotective and antidepressant properties of ligands of NPY receptors Y2R and Y5R, currently backed by the murine model [158]. A correlation between rats with anxiety-like behavior on levels of NPY, calcitonin gene-related peptide, and neurokinin has been reported by Carboni et al. 2022. A most recent study studied transcriptional and translational levels of NPY receptors in the prefrontal cortex and hippocampus of normal brains (control subjects) and suicidal subjects (study group) [159]. In both studied parts of the brain, a significant decrease in *NPY* mRNA and also upregulation of *Y1R* and *Y2R* mRNA was observed in the study group, along with a significant decrease in expression of the NPY protein in the prefrontal cortex of the study group [160].

### 2.9. Neuropeptide S

Neuropeptide S (NPS), a 20-amino acid neuropeptide, is implicated in sleep, arousal, feeding behavior, anxiety, and stress adaptation [19,20]. In a study using Flinders Sensitive Line (FSL) versus Flinders Resistant Line (FRL) rats, the intracerebral application of NPS resulted in reduced depression and anxiety-related behaviors in FSL animals, suggesting its anxiolytic and antidepressant role [161]. In humans, a polymorphism of the NPS receptor in the amygdala was identified in males with panic disorder [162]. The mechanism of the anxiolytic effect of NPS was examined in a study, which investigated the interactions between NPS and other mediators, namely noradrenaline, serotonin, glutamate, GABA, dopamine, and acetylcholine, using mouse frontal cortex synaptosomes labeled with radioactive neurotransmitters. They found out that NPS binds to neurons in the frontal cortex, reducing evoked serotonin and norepinephrine release [163]. A later review described the interplay between NPS and oxytocinergic systems [164]. More recent evidence supports the anxiolytic role of NPS. Tillmann et al. used an adeno-associated viral vector to induce the overexpression of NPS in rat amygdala. The NPS overexpression had a massive anxiolytic effect in rats. However, the study did not confirm the antidepressant properties of NPS [165].

### 2.10. Neuropeptide FF

Neuropeptide FF (NPFF), an 8-amino acid neuropeptide, belongs to RF-amide-related peptides—the RFRPs, which play a role in physiological processes, such as the modulation of morphine-induced analgesia, blood pressure elevation, and increased pancreatic somatostatin secretion [166,167]. NPFF utilizes two receptors, NPFFR1 and NPFFR2, which also represent new therapeutic targets [168]. NPFF and different RFRPs do have similar effects: they do share the NPFFR1; however, further effects in mammals are not well researched yet [169]. A connection with the HPA and HPG axis in relation to NPFF receptors has been studied. NPFFR2 has been found to activate the HPA axis and induce anxiety in rats and mice [170]. NPFF has a role in the reproductive system, the HPG axis, the autonomic nervous system, and also in pathways affecting stress response, food intake, and energy balance [171,172]. A recent study reported stronger negative feedback on the HPA axis (trend toward resistance) after a single prolonged stress event in NPFFR2 knockout mice [173]. The prevention of lipopolysaccharide-induced depressive-like responses in mice was achieved in NPFFR2 knockout mice [174].

### 2.11. The Galanin Family

Galanin (GAL) in humans is a 30-amino acid neuropeptide. In other species, it consists of 29 amino acids, and the mature forms are derived from a 123- or 124-peptide precursor for humans and other species, respectively [175]. The precursor peptide contains sequences for other galanin family members, the galanin-message-associated peptide (GMAP) and GAL (1–15), the galanin N-terminal fragment [176]. Other members of the galanin family are galanin-like peptide—GALP (encoded by a GALP gene)—and alarin (encoded by a splice variant of GALP) [177]. Galanin family members bind to specific GPCRs—GALR1, GALR2, and GALR3, of which, at least in rodents, R1s are mainly centrally expressed, and R2s and R3s are both in central and peripheral tissues [178]. The 1 and 3 receptors cause K+ efflux, while GALR2 elevates Ca2+; thus, their effects induce different pathways. Receptors 1 and 3 activate Gi-coupled inhibitory signaling, and GALR2 activates Gq-coupled inhibitory signaling [179]. GAL and GAL (1–15) exhibit a high affinity for type 1 and 2 receptors, and spexin (discussed earlier) has an affinity for type 2 and 3 [180]. GAL is a “multitalented peptide”, involved in a wide network of functions, such as the regulation of feeding, learning, and memory, as well as nociception and changes in mood and behavior, thus posing a potential therapeutic target for affective disorders or drug addiction [181,182,183]. GAL is expressed in many species, both in the central and peripheral nervous system, gastrointestinal tract, and endocrine organs, with effects reaching even processes such as the control of prolactin, growth hormone, and luteinizing hormone release [175,184]. The GAL regulation of glucose and energy homeostasis was studied in animal models, where GAL-KO mice were glucose intolerant; data suggests that GAL might also play an inhibitory role in insulin secretion on a neuronal level [185]. In mice, an increase in metabolism and energy usage is caused by the ΔFosB over-expression mediated by hypothalamic neurons, which need GAL as the inducer [186,187]. In the central nervous system, GAL is co-expressed with serotonin and norepinephrine, of which both are involved in depressive disorder [188]. GAL modulates the norepinephrine in the locus coeruleus and serotonin in DRN [188]. This effect is likely to occur because parts of the brainstem (e.g., DRN—dorsal raphe nucleus) neurons co-express serotonin (5-HT) and galanin [189]. Microinjections of 3 nmol galanin into the dorsal periaqueductal grey matter impaired learned anxiety in rats without changing unconditioned fear, suggesting an inhibition of acquisition of anxiety-like responses [190]. However, GALR1 and GALR2 were found to have an opposite effect on anxiety, being -genic vs. -lytic, respectively [191]. Studies have shown the effects of galanin on anxiety and depression; however, a correlation between depression and plasma levels of GAL proved positive only in women [192,193]. Results of animal studies show an upregulated GALR1 expression in the prefrontal cortex in a postnatal depression model, while 5-HT and 5-HIAA expression were downregulated [194]. A mice anxiety model with GALR3-KO mice reported elevated anxiety-like behavior and reluctance toward social behavior, which seems a result of GAL-mediated regulation, either through GAL (1–15) or spexin [195]. As the prefrontal cortex is an important part of the brain affecting mood and behavior, results suggest that galanin may play an important role. A possible connection between stress-related anxiety and the resulting depression lies in pain. In humans and animals alike, chronic pain leads to changes in mood and behavior and thus a coexisting feeling of anxiety and depression. As a response to inflammation and pain in rodents, expressions of type 1 and 2 receptors are upregulated in the nucleus accumbens, serving as a confirmation of the previous theory [196,197]. Recently, the genotyping of non-coding variants of galanin found a presence of a T allele at rs1042577 to be associated with greater levels of anxiety, while haplotype analysis pointed to a significant association of rs948854_C-rs4432027_C combination with anxiety [198].

Galanin (1–15) enhanced the antidepressant effects induced by the 5-HT1AR agonist 8-OH-DPAT in the forced swimming test [199], indicating a viable diagnostic and therapeutic path targeting the galanin-serotonin receptor interaction [200]. Tested by a proximity ligation assay, a possible heterocomplex of GALR1 and GALR2 in the dorsal hippocampus and DRN was proposed, as knockout (KO) of the first caused a disappearance of GAL (1–15) effect, while KO of the second caused a reduction of effect, possibly by dissipation of the receptor heterocomplex [201]. A different angle suggests that GAL has a higher potency of signaling activation via the GALR2, while GAL (1–15) has a high affinity to the GALR1-2 heteroreceptor, and this disbalance in signaling could lead to depression-like effects [202].

GALP is a 60-amino acid peptide with a homologous sequence to galanin that enables the binding to and activation of all three GAL receptors. GALP however, is almost exclusively expressed in ARC and the posterior pituitary [203]. To date, GALP has not been connected to anxiety or depression. Data suggest relations to metabolism, energy homeostasis, and reproduction; changes in these processes are observed only after acute (not chronic) stress exposure in rats [204,205]. Alarin does not bind to GAL receptors, which means its function is mediated through an alarin-specific receptor, which has not yet been discovered.

### 2.12. Spexin

Spexin (SPX) or neuropeptide Q (NPQ) is an endogenous neuropeptide consisting of 14 amino acids in its mature form—the same for humans and mice, but not rats [206]. SPX is produced in the pancreas and to date does not have a specific receptor; however, it was found to interact with galanin receptors GALR2 and GALR3180. The presence of SXP in CNS and endocrine, gastrointestinal, urinary, and reproductive organs has been recorded in multiple species, suggesting a wide array of pathways connected to this neuropeptide [207,208,209,210]. Spexin-based GALR2 agonist with increased stability created by D-Asn1 substitution produced an anxiolytic effect in mice after intracerebroventricular administration [211]. Studies have also linked spexin to the regulation of glycemia (by a reciprocal inhibitory relationship between insulin and spexin) and also to body weight regulation, as observed in a study comparing obese children to their average-weight peers [175,212]. A study on rats by Palasz et al. demonstrated that the prolonged intraperitoneal administration of escitalopram caused an upregulation of the SPX gene in the hippocampus and striatum, while expression in the hypothalamus was downregulated [213]. A subsequent study by the same team found that the prolonged intraperitoneal administration of haloperidol and chlorpromazine increased the SPX and proopiomelanocortin (POMC) mRNA expression in the rat amygdala, while the expression of kisspeptin-1 mRNA decreased [214]. All aforementioned results indicate an anxiolytic, antidepressant, and antipsychotic effect of SPX-related pathways in the amygdala [215].

Animal studies on the effects of spexin in anxiety and depression indicate a connection between SPX and the CRH system, while the CRH system has an established connection to the serotonergic system [216,217,218]. Collectively, in animal studies, the location of spexin-producing neurons is often in close proximity to the serotonergic (5-HT) neuron fibers, which complements theories on SPX-CRH-5-HT inter-actions and thus a possible effect of SPX on mood and behavior. A 3.5-fold increase in the local mRNA expression of CRH and a 30% lower SPX mRNA expression in mice extensively exposed to unpredictable stress was observed. This effect deepens by a CRH injection into the hippocampus-mediated by the CRHR2 [219]. Overexpression of SPX1 has an anxiolytic effect in transgenic zebrafish, which is mediated by galanin 2a and 2b orthologous receptors [220,221,222]. Fish under chronic social defeat stress had a 2.4-fold increase in plasma cortisol levels and upregulated expression of SPX1a and SPX1b paralogues in the optic tectum, hypothalamus, and mid-brain [223].

### 2.13. Kisspeptin

Kisspeptins are a family of four peptides (kisspeptin-10, 13, 14, and 54) cleaved from a 145-amino acid precursor which interacts with the KISS1 receptor also known as the GPR54 or metastin receptor [224,225]. Kisspeptins play roles in multiple non-hormonal and non-mood-related pathologies ranging from pulmonary fibrosis and amyloid-beta pathology to breast cancer or metastasis process [226,227,228,229,230]. In the hormonal-related area, researchers observed effects on the stimulation of GnRH secretion, regulation of energy balance, and the effect on fat tissue or the beta cells in the pancreas [231,232,233,234,235,236]. These connections to puberty and the reproductive system are caused by interaction with the HPG axis, which indicates a two-way relation with energetic metabolism not limited only to developmental stages but also later in life. Proof of this theory was observed in postmenopausal women and ovariectomized monkeys with the highest concentration of kisspeptin located in the hypothalamic infundibular nucleus, which is due to hypertrophy and upregulated KISS1 gene expression, reversible with estrogen administration [237]. High concentration locations in rodents are the anteroventral periventricular nucleus and the arcuate nucleus [238]. In rodents, expression in the amygdala has been observed with induced expression in puberty stimulated either by sex-related steroid hormones or just going independently on par with their secretion [239].

Questions about KISS1 and its relation to mood or social behavior, resulting in anxiety and depression, are only recently being addressed, even though stress-induced increases in adrenal glucocorticoids are known to suppress the HPG axis, and reproductive dysfunctions lead to mood changes in humans. Even without psychological effects, GnRH affects mood and regulates social behavior through the HPG axis in humans and rodents alike [240,241,242,243]. Furthermore, a stress-induced desire for high-sugar foods affects the energetic metabolism, thus over time resulting in obesity and, again, in humans, to anxiety-like or depressive-like behavior. Hofmann et al. positively correlated kisspeptin with BMI and body fat mass in patients with anorexia nervosa; however, no relation to anxiety or depression was observed [244]. Kisspeptin-13 stimulates the HPA axis by interaction with α2 receptors where intracerebroventricular administration induces hyperthermia and causes anxiety-like behavior in rats, while interaction with 5-HT2 receptors causes antidepressant-like effects in mice [245,246].

### 2.14. Substance P

Substance P (SP) is an 11-amino acid neuropeptide of the tachykinin family that comprises neurokinins A and B along with hemokinin-1. Tachykinins share three neurokinin receptors, the NK-1R, NK-2R, and NK-3R (also known as TACR1; TACR2; and TACR3). Substance P is present in the central nervous system and primarily secreted in the amygdala, hippocampus, or basal forebrain; however, cells connected to the inflammatory processes also exhibit the capacity for its secretion. Substance P has been found to interact with the serotonergic, dopaminergic, and noradrenergic systems, which connects it to many biological processes such as stress regulation, nociception, or homeostasis regulation [247]. As these systems also play a major role in the pathophysiology of depression, a potential role of Substance P as a co-factor and biomarker in depressive disorders and anxiety is suggested. It is supposed to be involved in the activation of the sympathetic system and HPA axis in response to stressors [248]. This is supported by the observation that the central administration of Substance P and NK-1R (neurokinin type 1 receptor) leads to depression-like and anxious behaviors in animals, while NK-1R antagonists cause a decrease in c-Fos expression in PVN along with an anxiolytic effect [249,250]. In humans with major depression, increased serum levels of substance P were identified, compared to healthy controls, and after treatment, a reduction in serum Substance P levels was observed [247]. The NK-1R antagonists have been demonstrated to have antidepressant effects, and they also reduce neuronal activity in the brain areas involved in the response to stress [251]. In rats, stress induction was accompanied by the release of Substance P in the medial amygdala and induced anxiety-like behavior antagonizable by the receptor antagonists [252]. However, a recent study on the availability of the NK-1R in patients suffering from major depressive disorder did not find any correlation in baseline expression compared to healthy controls [253]. Even though NK-1R receptor antagonists were primarily created as antidepressants showing sometimes mixed, but mostly promising, results in the treatment of depression and anxiety, research on these antagonists reported various concurrent effects, of which antiemetic and antipruritic effects were further studied [254,255,256]. Upregulation of the tachykinergic system (Substance P included) has been seen in skin biopsies of patients with atopic dermatitis [257]. Advancements in pain management with the use of capsaicin or β-caryophyllene (BCP) were also reported [258,259,260].

### 2.15. Neurotensin

Neurotensin (NT), a 13-amino acid neuropeptide, is present in the bed nucleus of the stria terminalis, which is a structure supposed to be involved in anxiety. Its potential role in the pathophysiology of anxiety is so far insufficiently understood. NT in rat brains has been found to be associated with dopaminergic and glutamatergic systems [261,262]. In experiments, neurotensin has been found to increase the transmission in the bed nucleus of the stria terminalis together with corticotropin-releasing factor, whereas a blockade of NT receptors prevented stress-related anxiety-like behaviors [263].

NTSR1 (neurotensin receptor 1) knockout mice showed increased despair and anxiety. Additionally in dark phases, they spent a lower percentage of time in REM sleep [264]. The NTSR2 (neurotensin receptor 2) was marked as one of the viable targets for fear-inhibiting neural pathways [265]. In a study on patients with acne vulgaris and their quality of life, anxiety, and depression, neurotensin levels were significantly higher in the study group compared to controls [266]. NT and xenin in obese patients displayed univariable concentrations in males and females; however, females exhibited higher psychometric values, as well as a positive correlation with stress, anxiety, depression, and eating disorders, but men did not [267].

### 2.16. Hypocretins

Hypocretins are neuropeptides originating in the lateral hypothalamus, first discovered in 1998. Hypocretin neurons provide interactions within the whole central nervous system, interact with other neurotransmitters, and activate the HPA axis. The hypocretin system consists of two neuropeptides, the 33-amino acid Orexin-A (OrxA = Hcrt-1) and the 28-amino acid Orexin-B (OrxB = Hcrt-2), and two types of receptors, the HCRT-R1 and the HCRT-R2. Hypocretins are involved in the regulation of arousal, homeostasis, and circadian rhythms, which if dysregulated lead to daytime drowsiness and even narcolepsy [268,269,270,271]. Orexin type 1 receptors have been associated with the promotion of anxiety and depressive-like behavior, while the type 2 receptors exhibit anxiolytic and antidepressant effects [272]. HCRT-2R knockout enhances anxiety and depression [272].

Their potential role in anxiety via the HPA axis activation was investigated shortly after their discovery on animal models when Kuru et al. had centrally administered both orexins into rat brains, which led to an increase in plasma levels of ACTH presumably via CRH and arginine vasopressin release and c-Fos gene induction [273]. This suggestion was later supported by the observation that humans with panic disorder had increased orexin levels in cerebrospinal fluid compared to controls [274]. A reciprocal relationship with CRH is suggested, as neurons of both neuropeptides are innervating regions in the limbic system, especially regions associated with anxiety and depression. The connection to the HPA axis leads us to believe that in reaction to stress, the orexin system increases as a reactive and coping mechanism for animals in threat adaptation. Dysregulation in the adaptation process and hyperactivation of the HPA axis is a potential connection to anxiety and depression. Additionally, the potential role of orexinergic neurons in depression was described in rodents, when unpredictable chronic mild stress led to the increased activity of orexinergic neurons and reduced expression of HCRT type 2 receptor, but this effect was reversed by the treatment with antidepressant agent fluoxetine [275]. More recently, a contradictory finding was reported in an animal study, in which intracerebral application of an orexin receptor agonist resulted in a reduction of anxious and depressive behaviors, whilst its antagonists induced anxious and depressive behavior [268]. The latest reviews suggest different roles of orexins in anxiety and depression, concluding that orexins are anxiogenic, while they also probably play a role in depression, but this role is to be further elucidated [271,276].

### 2.17. Phoenixin

Phoenixin (PNX) exists in two isoforms: PNX-14 and PNX-20, which consist of 14 and 20 amino acids, respectively. Both are coupled with G-protein-coupled receptor-173 (GPR173) [277]. Jiang et al. described its dose-dependent anxiolytic-like behavior in a male Kunming strain of Swiss mice [278]. PNX has a connection to the HPG axis and stimulates the GnRH-stimulated LH release from pituitary cells [279]. A similar effect was observed in the open field test, which is often used for testing anxiety-like behavior and locomotion [280]. Prinz et al. mapped the expression sites of the 14-amino acid phoenixin in the brain and peripheral tissues of Sprague Dawley rats using a specific phoenixin 14 antibody and found a high density of phoenixin 14 reactivity in the central amygdaloid nucleus, which might be involved in its anxiolytic activity [281]. Later, in research using restraint stress in Sprague Dawley rats, Friedrich et al. were able to prove that phoenixin expression in the dorsal motor nucleus of the vagus nerve, raphe pallidus, and nucleus of the solitary tract positively correlated with C-fos expression in these nuclei in response to restraint stress [282]. A recent review also suggests that PNX modulates the HPA axis during chronic stress [277]. There are also human studies indicating the role of phoenixin-14 in the modulation of anxiety. For instance, a study performed on 68 male obese inpatients who were treated for obesity and associated comorbidities showed that phoenixin is negatively associated with self-reported anxiety as assessed using the Generalized Anxiety Disorder-7 form [283].

### 2.18. Relaxin-3

Relaxin-3 (RLN-3) is a newly discovered 51-amino acid neuropeptide. Primary research in the past has been performed on rats with a recent inclusion in human research. This neuropeptide is mainly located in the nucleus incertus (NI), with efferent ways within the amygdala [284,285]. The relaxin-3 receptor (RXFP3) is located in the brain and has been linked to stress, arousal, and feeding [286,287]. Relaxin-3 knockout mice showed similar results to relaxin-3 knockout (KO) rats in the past, where no presence of the protein in brain tissue was found compared to wild-type (WT) mice [288]. Relaxin-3 neurons located in the midbrain, and pons were associated with dysfunctional modalities (e.g., stress, arousal, anxiety) in neuropsychiatric diseases [289]. A murine model on relaxin-3 KO mice did not show differences in depression-like behavior, social interactions, or fear conditioning, but showed a slight increase in anxiety-like behavior compared to WT mice [290]. However, a later murine model testing an RXFP3 agonist demonstrated decreased cumulative neurogenic stressors and anxiety-like behavior, namely in mice with former exposure to anxiety tests [291]. RLN-3 KO mice were also hypoactive during the dark/active phase and on home-cage running wheels [292,293]. RLN-3 and related pathways, via a stress-induced increase in mRNA expression, are connected to a stress-related increase in food intake, especially high-sugar foods, with higher effects in females [294,295]. This behavior is decreased with RXFP3-antagonists [296]. Habitual eating over time can lead to further obesity and metabolic disorders that even deepen the stress and anxiety-related symptoms [297]. RLN-3/RXFP3 deficiency did not prove effective in altering anxiety, anhedonia, and social interactions in a model of cessation of exposure after chronic methamphetamine administration [298]. Recently, therapy of anxiety, depression, and related disorders in rats was targeting the RLN-3/RXFP3 system via intranasal delivery of an RLN-3 mimetic with positive results [299]. A novel study on female rats with mRNA-induced RLN-3 depletion led to alterations in food intake, a 2% decrease in body weight, and increase in anxiety-like behavior, although only in a large open field but not in an elevated plus-maze or light/dark box [300]. RLN-3-depleted rats had also disrupted corticosterone regulation and increased oxytocin and AVP, but not CRH, which indicates more of a fine-tuning effect of RLN3 neurons on stress, food intake regulation, and neuroendocrine responses [300]. In a systematic review, Wong et al., 2021 concluded that there is evidence in the literature on the RLN-3/RXFP3 to promote arousal and suppress anxiety- and depression-like behavior, but this field lacks high-quality clinical studies [301]. A most recent study associated the RLN-3/RXFP3 system as an important factor in molecular aging and multiple forms of aging-associated diseases [302].

### 2.19. Nesfatin-1

Nesfatin-1 a novel neuropeptide made of 82 amino acids, is cleaved from its precursor nucleobindin 2 (NUCB2) and was first discovered as an anorexigenic protein in rat hypothalamus [303]. Expressions are located predominantly in the brainstem, but also in the forebrain, hindbrain, and spinal cord [304,305]. In female rats, activity-induced anorexia and restricted feeding resulted in elevated expression of nesfatin-1 in the hypothalamic nuclei, locus coeruleus, and the rostral part of the solitary tract, indicating a possible role in the pathology of anorexia and stress-related mood changes [306]. The sex-specific expression of NUCB2 mRNA compared to controls is 1.8-fold elevated in male suicide victims, however 2.7-fold decreased in female suicide victims [307]. In obese subjects, the female gender is more associated with nesfatin-1 elevation than males, especially in perceived stress, anxiety, and depressiveness, but does not necessarily correlate with BMI [308,309,310]. Although in an inpatient setting after treatment and improved self-evaluated anxiety scores, levels of nesfatin-1 did not change [311]. Positive correlations in females and even inverse correlations in males could indicate different reactions to cope with stress in both sexes, and slow changes in expression levels do indicate a coping mechanism. However, a question stands what results would show a comparison to human subjects without obesity, if the correlation is caused rather by effect than by being the cause. Normal weight rats, if intracerebroventricularly administered doses of the protein, exhibited increases in anxiety, depression-like behavior, and anhedonia [312]. Not directly addressed, but in a study on female patients with fibromyalgia syndrome (FMS), subjects and controls were in the same weight/BMI category (+4 kg median weight in subjects), and measured serum nesfatin-1 concentrations were higher in subjects with FMS and anxiety [313]. These results could point to the nesfatin expression levels elevated in hypersensitive patients mediated through the brain–gut axis (visceral sensitivity) or while feeling discomfort and pain mediated through CRH neurons [314,315]. Suppression of this elevation has anxiolytic effects but does not prevent the effect of the CRH system [316]. Nesfatin-1 levels in subjects with antisocial personality disorder chosen upon involvement in at least one crime incident demonstrated higher impulsivity and aggression, while expression levels of nesfatin-1 in serum were lower than in controls [317]. In relation to the first depressive episode in adolescents, nesfatin-1 levels were lower in the study group equally in both sexes [318]. Alcohol dependency in patients in their first month of abstinence was not correlated with nestafin-1; however, the scores on the self-rating depression scale were [319]. These results indicate that mood, anxiety, and depression are the targets for nesfatin-1 and not addiction. The connection of nesfatin-1 to the HPG axis through CRH upregulation could be a possible pathway to mood and behavior changes [320].

### 2.20. Nociceptin

Nociceptin (Orphanin FQ—N/OFQ) is a 17-amino acid neuropeptide that has been studied in association with body weight regulation, emotions, stress, anxiety, depression, and substance dependence [321,322,323]. High expression levels of the nociceptin receptor (NOP or opioid receptor-like 1—OPRL1) have been recorded in the limbic system, thus the aforementioned associations [324,325]. ORL1 receptors are located on serotonergic neurons in the dorsal raphe nucleus, suggesting an anti-depressive/anxiolytic effect through this pathway [326]. In-escapable stress situations stimulate N/OFQ expression, inducing stress and anxiogenic behavior. This effect is modulated by the OPRL1 antagonist in male rats [327]. N/OPQ antagonist efficacy in controlling depression symptoms proved positive in a pre-clinical rodent model. In humans, these effects were lesser, although positive in total compared to placebo [328]. The central administration of OPRL1 antagonists induces anxiolytic effects in mice [329]. However, subsequent studies reported OPRL1 agonists inducing aggressive behavior, while antagonists prevented this behavior in mice [330]. Only a marginal relevance of N/OPQ in aversive learning in mice was recorded, although memory impairment is suggested [331,332]. New data complements the claim of functional heterogeneity in these receptors [333,334]. The OPRL1 system is suggested a standalone of other opioid receptors, as no compensatory induction was recorded in N/OPQ knockout rats [335]. A connection to the HPA axis and related neuropathic pain shows the wide utilization of the OPRL1 receptors, affecting even stress-related mechanisms in patients with post-traumatic stress disorder and traumatic brain injury [336,337].

### 2.21. Cholecystokinin

As the name suggests, cholecystokinin (CCK) was first discovered in the gastrointestinal tract as a digestive peptide hormone that participates in the processing of fats and proteins by stimulating the pancreas and gallbladder [338]. Later, CCK was also described as a neurotransmitter found in many areas of the mammalian brain, including the limbic system [339]. CCK is of a variable length, depending on the posttranslational modification of preprocholecystokinin, the 150-amino acid precursor [340]. The expression of CCK was found to be ubiquitous, making it one of the most widely expressed neuropeptides in the brain [341]. The role of CCK as an anxiogenic substance was studied in both human and animal studies. In 1991, Bradwejn et al. performed a clinical trial with the main aim to study CCK tetrapeptide (CCK-4) activity in patients with panic disorder. Upon injection of 50 micrograms of CKK-4, 12 out of 12 patients in the experimental group experienced a panic attack. Interestingly, 7 out of 15 subjects in the control group had similar symptoms [342]. By the end of the 1990s, another clinical trial on human subjects investigated the CCK concentration in the cerebrospinal fluid (CSF). The authors found an inverse correlation between CSF levels of CCK and proneness to anxiety, depression, and suicidal behavior [343]. In a knock-down mouse model, Del Boca et al. concluded that the experimental reduction of CCK mainly in the basolateral amygdala produced anxiolytic and antidepressant-like effects [344]. CCK activity has been also linked to social stress-induced anxiety and depression. Establishing a mouse model of social defeat stress, Vialou et al. demonstrated the overexpression of ΔFosB in model animals. ΔFosB is a transcription factor promoting stress susceptibility, whose signaling was mediated via the CCK-B receptor. The role of CCK-ergic neurotransmission in the pathogenesis of anxiety and depression was also evidenced by the infusion of CCK-B receptor agonists, producing similar effects [345]. The correlation between CCK-4 levels and anxiety and depression was shown repeatedly in a mouse model. Based on the known action of NPY as the substance with virtually opposing effects on anxiety and depression, Desai et al. highlighted that NPY and CCK-4 attenuate each other upon experimental intervention, producing the respective effects on the parameters of anxiety and depression [346]. Despite all the aforementioned, as of today, no clinical trial has been able to show that CCK-B receptor antagonists have clinically significant effects in humans in terms of diminishing the symptoms of anxiety and depression [347]. Moreover, the authors also discovered that the CCK role in anxiety and depression is far more complex. Whereas the acute administration of CCK-B receptor antagonist Ly225.910 to mice was indeed anxiolytic, the prolonged carry-over effect was different, causing reluctance to perform a given task. Surprisingly, the endogenous CCK reactivation of the CCK-B receptor restored the benchmark anxiety levels [347]. In summary, CCK is a promising target in the treatment of anxiety and depression, though a more robust body of clinical trials has to be at hand to implement this approach to the evidence-based complex management of these two conditions [348].

### 2.22. Other Neuropeptides

Calcitonin gene-related peptide (CGRP) is a 37-amino acid neuropeptide that exists in two forms (alpha and beta) and binds to the CGRP receptor [349]. The intracerebroventricular administration of CGRP receptor antagonists and CGRP resulted in increased locomotor activity, and decreased depression-like behavior, but failed to show effects in anxiety-oriented tests [350]. The administration of CGRP to adult mice resulted in increased depression-like behavior that could be reversed by the receptor antagonists [351]. CGRP is linked to immunity and the gut–brain axis with antimicrobial and antifungal properties [352].

Neurokinin A (also known as Tachykinin 1 or substance K) is a 10-amino acid neuropeptide encoded by the *TAC1* gene that belongs to the tachykinin family and binds to tachykinin receptors. The deletion of the *TAC1* gene results in decreased anxiety-like and depression-like behaviors in mice [353]. As in the case of Substance P, NK-1R and NK-2R receptor antagonists decrease anxiety and depressive-like behavior in rats [354]. A recent study reported significant correlations between CGRP, NKA, NPY, NTS, and anxiety-like behavior in rats [159]. Elevated levels of NPY in the cingulate cortex and striatum, while CGRP was elevated in the frontal cortex and hippocampus and NKA in the entorhinal cortex [159].

## 3. Conclusions

Anxiety and depression have been accompanying humanity and animals throughout history. Mental health was a big unknown for a long time and even nowadays, with regiments of medical procedures or drugs, we are unable to cure all patients, or even achieve long-lasting remission in many. The principle of medical practice is to improve the quality of life for patients. However, in patients with mental disorders, this is not an easy feat. As these disorders have a very variable clinical presentation, we often misinterpret some symptoms or miss them entirely. In these cases, an examinable, measurable level of some molecule or molecules could greatly improve the often-abstract symptomatology of psychiatric disorders. Often, pain or fear leads to anxiety or depression, emotional disbalance, and slowly even to physical deterioration, starting from benign symptoms, through hair loss, digestive problems, and lack of appetite, but it can even lead to suicide.

The world of neuropeptides is vast, and addressing every single one would be no easy feat. In this review, we have taken a close look at this topic, and we can conclude that, based on available data, some neuropeptide families and systems are clearly associated with both anxiety and depression. These groups include the melanocortins, the CRH system, and the galanin family. Individual neuropeptides with effect on both disorders are NPY, SPX, Substance P, and CCK. Targeting these molecules and their receptors will improve our diagnostic capability, and therefore, our targets of treatment will be more focused and hopefully effective. Understanding these pathways and using their receptor agonists or antagonists could prove an irreplaceable tool for treating many disorders, not only mental; as we have seen, energy metabolism, reproduction, and many other physiological functions are also mediated or affected by neuropeptides.

Decades into research and several pathways are yet only partially uncovered and their reciprocal regulation not thoroughly understood. As changes in mediation between species exist, we have to overcome these obstacles and move the research forward. So, in the end, we praise all researchers who have contributed to the enormous effort in the research of neuropeptides and the continuation of this research; especially, clinical research in the future will play a pivotal role in understanding mental disorders such as anxiety and depression, revealing their etiology and pathogenesis, and improve upon treatment.

## Figures and Tables

**Figure 1 behavsci-12-00262-f001:**
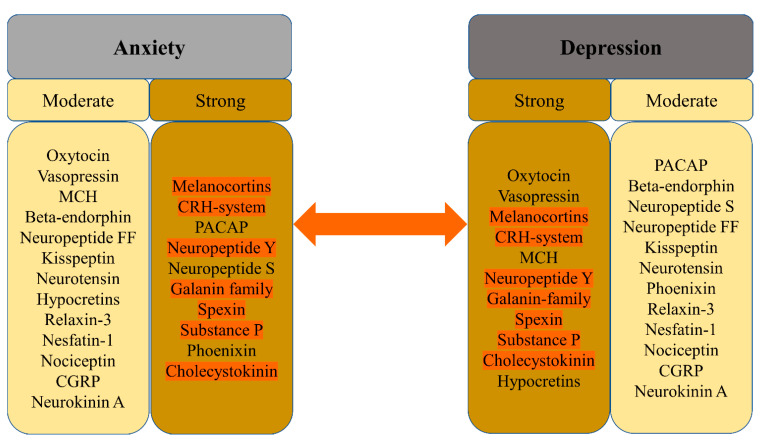
Association of neuropeptides with anxiety and depression based on evidence in research.

**Figure 2 behavsci-12-00262-f002:**
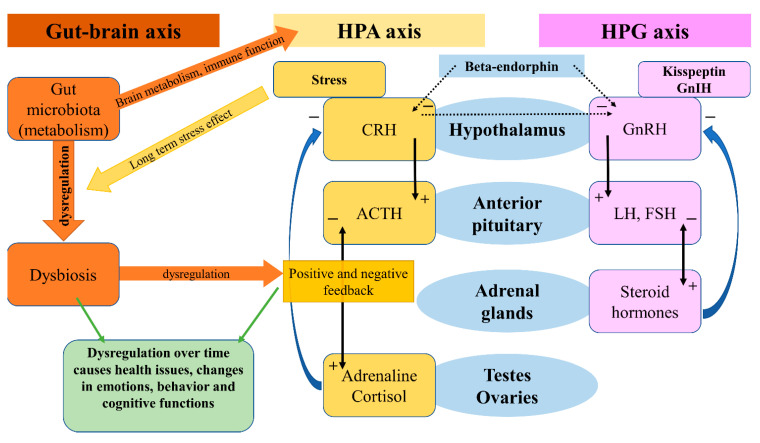
Gut-brain axis, HPA axis, HPG axis and their interactions.

**Table 1 behavsci-12-00262-t001:** Anxiety and depression-related neuropeptides, encoding genes, and related receptors. (Receptors of neuropeptides without any relation to anxiety or depression are not mentioned in this table.)

Neuropeptide	Gene (Symbol)	Cytogenetic Location	Related Receptors
Oxytocin	*OXYTOCIN (OXT)*	20p13	OXTR
Vasopressin	*ARGININE VASOPRESSIN (AVP)*	20p13	AVPR1A; AVPR1B; AVPR2
Adrenocorticotropic hormone	*PROOPIOMELANOCORTIN (POMC)*	2q23.3	MCR1; MCR2; MCR3; MCR4; MCR5
Corticotropin-releasing hormone	*CORTICOTROPIN-RELEASING HORMONE (CRH)*	8q13.1	CRHR1; CRHR2
Urocortin 1	*UROCORTIN (UCN)*	2p23.3	
Urocortin 2	*UROCORTIN II (UCN2)*	3p21.31	
Urocortin 3	*UROCORTIN III (UCN3)*	10p15.1	
Pituitary adenylate cyclase-activating polypeptide	*ADENYLATE CYCLASE-ACTIVATING* *POLYPEPTIDE 1 (ADCYAP1)*	18p11.32	ADCYAP1R1; VIPR1
Melanocyte stimulating hormone	*PROOPIOMELANOCORTIN (POMC)*	2p23.3	MC3R; MC4R
Melanin-concentrating hormone	*PRO-MELANIN-CONCENTRATING* *HORMONE (PMCH)*	12q23.2	MCH-R1; MCH-R2
Beta-endorphin	*PROOPIOMELANOCORTIN (POMC)*	2q23.3	OPRM1 other μ-opioid receptors
Neuropeptide Y	*NEUROPEPTIDE Y (NPY)*	7p15.3	NPY1R; NPY2R; NPY5R
Neuropeptide S	*NEUROPEPTIDE S (NPS)*	10q26.2	NPSR1
Neuropeptide FF	*NEUROPEPTIDE FF-AMIDE PEPTIDE (NPFF)*	12q13.13	NPFFR1; NPFFR2
Galanin	*GALANIN (GAL)*	11q13.2	GALR1; GALR2; GALR3; GPR151
Galanin-like peptide	*GALANIN-LIKE PEPTIDE (GALP)*	19q13.43	GALR1; GALR2
Spexin	*SPEXIN HORMONE (SPX)*	12p12.1	GALR2; GALR3
Kisspeptin	*KISS1 METASTASIS SUPPRESSOR* *(KISS1)*	1q32.1	KISS1R
Substance P	*TACHYKININ 1 (TAC1)*	7q21.3	TACR1; TACR2; TACR3
Neurotensin	*NEUROTENSIN (NTS)*	12q21.31	NTSR1; NTSR2
Hypocretin	*HYPOCRETIN (HCRT)*	17q21.2	HCRTR1; HCRTR2
Phoenixin	*SMALL INTEGRAL MEMBRANE PROTEIN-20 (SMIM20)*	4p15.2	GPR17
Relaxin 3	*RELAXIN 3 (RLN3)*	19p13.12	RXFP3
Nesfatin-1	*NUCLEOBINDIN 2 (NUCB2)*	11p15.1	Not discovered
Nociceptin	*PREPRONOCICEPTIN (PNOC)*	8p21.1	OPRL-1
Cholecystokinin	*CHOLECYSTOKININ (CCK)*	3p22.1	CCKAR; CCKBR
Calcitonin gene-related peptide	*CGRP RECEPTOR COMPONENT (CGRP)*	7q11.21	CGRPR
Neurokinin A	*TACHYKININ 1 (TAC1)*	7q21.3	TACR1; TACR2; TACR3

## Data Availability

Not applicable.

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
