# Peer review of "Anxiety and Depression: What Do We Know of Neuropeptides?"

_behavsci, 2022, doi:10.3390/bs12080262_

Round 1

Reviewer 1 Report

In my opinion, the title is very attractive, but then the reader remain a bit disappointed regarding the main 'focus' of this review: neuropeptides. While in the abstract the authors talk about pathways related to the various neuropeptides, in the associated chapters we do not see many details on specific pathways involved in the mechanisms of action of the related drugs. In few word, I would have expected a more detailed description of each peptidergic system, with receptors and mechanisms involved. Only the 'HPA' axis is frequently cited but, however, it was not carefully described in the main text (maybe a graph or an illustration describing it will be useful). In addition, in the introduction there is a long digression relating to Covid19, instead of a widen and more detailed introductory part on neuropeptides in general, that are the protagonist of this review. I am aware that it is not an easy job, but I would try to be more focused in the introductory part and more detailed in each chapter. Some more diagrams or figures associated with the body of the text would not hurt.

Some sentences are left in the middle or there are punctuation errors and English should be revised.

Author Response

First of all, we thank the reviewer for the kind words and evaluation of our review. We also thank you for the constructive comments.

  1. In my opinion, the title is very attractive, but then the reader remain a bit disappointed regarding the main 'focus' of this review: neuropeptides. While in the abstract the authors talk about pathways related to the various neuropeptides, in the associated chapters we do not see many details on specific pathways involved in the mechanisms of action of the related drugs.

 Our review was not meant to focus on many specific drugs affecting pathways/receptors of reviewed neuropeptides, we wanted to focus on the associations between neuropeptides and their receptors and what/how they affect. We reworked the text and added figures according to the reviewers’ propositions, hopefully, this will make the text more of what it should be. Also, the size of the text proved to be quite larger than we first anticipated…

  1. In few word, I would have expected a more detailed description of each peptidergic system, with receptors and mechanisms involved. Only the 'HPA' axis is frequently cited but, however, it was not carefully described in the main text (maybe a graph or an illustration describing it will be useful).

 We tried to address the effects of these systems with the figure added to section 2, where the effects of the mentioned axes are depicted. Sadly, the HPG axis connection to neuropeptides is less studied than the effects of the HPA axis. The HPG introductory part has been enlarged. A connection HPG-NPY has been added. Now there is also added the gut brain axis is, but only marginally to make the introductory part more focused on neuropeptides.

  1. In addition, in the introduction there is a long digression relating to Covid19, instead of a widen and more detailed introductory part on neuropeptides in general, that are the protagonist of this review. I am aware that it is not an easy job, but I would try to be more focused in the introductory part and more detailed in each chapter. Some more diagrams or figures associated with the body of the text would not hurt.

 The introductory part has been reworked to include more about neuropeptides. Figure containing thHPA, HPG, and GBA axes have been added to the text.

  1. Some sentences are left in the middle or there are punctuation errors and English should be revised.

 The whole text has been revised and spelling errors in the text have been revised.

Reviewer 2 Report

This review discusses the roles of neuropeptides involved in anxiety and depression. It is an interesting review and well written. But I still have some comments on it:

1.     In the lines 33 to 39, the authors introduced the uncertain relationship between smoking habits, the microbiome and anxiety/ depression, and stress that neuropeptides play a role in substance dependence. However, I cannot understand the idea the authors want to convey, whether it would like to highlight the role of neuropeptides in smoking or microbe-mediated anxiety and depression, I think more references should be given to make the logic clear.

2.     In the introduction part, I think at least a brief introduction to neuropeptides should be given for the reason that this is a keyword of the review.

3.     In the lines 94 and 95, when giving CRH and ACTH for the first time, the full name should be given.

4.     In the 2.1. part, the authors introduced HPA and HPG, which may be important in anxiety and depression, however it is a little obtrusive to form a separate part, and I think it can be integrated in other parts.

5.     In the 2.2. part, the authors proposed the paradoxical role of OXT in anxiety and depression. I think it is necessary to explain or look at the mechanism that produces this paradoxical phenomenon, which can help us better understand the role of OXT.

6.     Some writing conventions should be noted, for example in the lines 427 and 428, the writing about mRNA should be NPY, Y1R and Y2R mRNA rather than NPY, Y1R and Y2R mRNA.

7.     In 2.7., 2.15. and 2.17. parts, the review of the progress of MCH, substance P and orexin in anxiety and depression is limited, the role of them is very important and should be reviewed more comprehensively.

8.     In this review, the authors provided a good overview of the role of neuropeptides in anxiety and depression, however, the neurobiological or molecular mechanisms seem to be overlooked, which is the first step to understand the role of neuropeptides. It is necessary to summarize this aspect, which will make this review more complete.

9.     In the conclusion part, a better summary of the full text is necessary, such as which types of neuropeptides are more important in anxiety and depression, what means can be used to study the effects of neuropeptides in the future, and the therapeutic potential of targeting their receptors, etc. Readers will get better inspiration form the summary of these aspects.

10.  For some neuropeptides with clear roles in anxiety and depression, it may be better if a schematic diagram of the role be given.

Author Response

First of all, we thank the reviewer for the kind words and evaluation of our review. We tried our best to address all comments:

  1. This section of the introduction has been reworked for clarity of logic, also in the second Section, 2.22 a short part for Other neuropeptides has been added.
  2. A brief introduction of neuropeptides has been added
  3. The full name of ACTH has been added. Now, however, CRH has been also mentioned in the introduction by full name, so in this part, the shortcut stays, duplicate of the full name will only be added in the individual section of CRH.
  4. We integrated the HPA/HPG part into section 2. with a little change to the first paragraph. Oxytocin continues as 2.1 – if it's not a problem for the reviewer we would rather keep the short reworked introductory section of HPA/HPG at the beginning of the neuropeptide section, as we aimed it as a brief introduction to the affected axes.
  5. Mention of the oxytocin paradox along with the citation has been added to the article.
  6. The proposed mRNAs have been changed to the italic font.
  7. All mentioned sections have been revised and information added. MCH and Substance P has also been mentioned in Figure 1 as having a strong association with both anxiety and depression, as was the Hypocretin association to depression.
  8. Comment on neurobiological and molecular mechanisms has been added to the introduction.
  9. The conclusion has been updated with more information from the article.
  10. A figure has been added to the introduction part addressing these associations.

Reviewer 3 Report

The authors well elaborated on neuropeptides and Anxiety and depression.

It would be great to show the single shot of summary figure.

So, readers quickly can get the contents of the review.

Author Response

We thank the reviewer for the kind evaluation of our review and for the constructive suggestion. A summarizing figure has been added to the Introduction – Figure 1. where we address the association of individual neuropeptides with anxiety and depression.

Also, a spelling check has been performed.

Round 2

Reviewer 1 Report

The aim of this review is to explore and better understand the role of different neuropeptides in anxiety and depression. 

Overall, the topic is really modern and attractive, with relevance for research and clinical studies.

However, the manuscript appears to me too much dispersed, unrelated, and complicated. I think that it can be very difficult for a reader to maintain the attention and I believe that this work, despite some interesting and innovative ideas, still needs major revision and a tough rearrangement work by the authors. 

I realize that this work is very long and complex, difficult to assemble, but at the same time it should reach a certain degree of simplicity and fluency in order to be then adequately considered and cited.

These are some of my suggestions:

Abstract

Line12.  My suggestion: and, in the worst case, to suicides.

Intro

Line 30: Different studies are aimed at different individual symptoms, using different methods in the evaluation of objective and subjective symptoms. These disorders exhibit gender and occupational differences and higher strain in patients suffering from serious diseases [1-3]

My suggestion: Several studies are aimed at addressing a multitude of individual objective and subjective symptoms, using diverse methods. These disorders exhibit indeed gender and occupational differences and higher strength in patients suffering from serious diseases [1-3].

Line 33: Increased susceptibility to substance abuse affected by coping mechanisms in anxiety and depression is often linked to nicotinism, alcoholism, or drug misuse and addiction.

My suggestion: Substances of abuse are often used to cope with anxiety and depression, leading to increased susceptibility to nicotinism, alcoholism, or drug misuse and addiction.

Line 38. suggestion: over time they may rather have opposing effects

(to avoid repeating the word these)

Line 39: is frequently seen in the clinical practice

Line 40: a recent systematic review that collected 140 studies concerning the association of cigarette smoking behavior with depression and anxiety reported contrasting results

Line 42: Many pathways with an involvement in anxiety…

Line 45: Both anxiety and depressive disorders

Line 49: To date, some specific genotypes, state and flexibility of the cardiovascular system, and function of the anterior cingulate cortex have the ability to change the outcome of treatment ( citations?).

Comment: Maybe better state and flexibility of the autonomic system? Only the function of the anterior cingulate cortex? And what about the prefrontal?

Line51: Research on neurotransmitters shows a contribution of the gamma-aminobutyric acid (GABA), serotonergic, and noradrenergic systems  (citations?)..

Comment: only these 3? And the dopaminergic? (see the monoaminergic hypothesis of depression) 

Suggestion: maybe it’s enough to add the word mainly, or a major contribution

Line54: Here there is a clear separation of topics that clashes and makes the text not very fluid. I would add a sentence that create a link between what you have said until now and the neuropeptides listed in the table. 

In Figure 1 are listed the 22 neuropeptides…. Sounds more elegant

Comment: It would be appropriate to indicate all the references that allowed the authors to make the table in Fig 1

From line 57 to 63: long and disconnected chapter I would move it at the end and reformulate

The whole introduction should be reformulated: I would start with the systematic data and then introduce the neuropeptides, that are the topic of the following chapter.  

comment: In table 1 are listed the neuropeptides with their precursor genes and receptors, but for some of them more receptors exist. Was it a choice of the authors to focus the attention only on the receptors involved in anxiety and depression and omit the ones which don't have an exact or important function? If so, it should be specified. eg: for this neuropeptide a number of receptors exist, but just these few are known to play a role.

Author Response

We thank the reviewer for constructive comments, and suggestions to further improve the stylistic part. As the review process is meant to improve on the original article, we are glad to be able to make these improvements.

The abstract and intro have been edited according to the reviewers’ suggestions.

Line 49: To date, some specific genotypes, state and flexibility of the cardiovascular system, and function of the anterior cingulate cortex have the ability to change the outcome of treatment ( citations?).

-citation has been added and sentence reformulated

Comment: Maybe better state and flexibility of the autonomic system? Only the function of the anterior cingulate cortex? And what about the prefrontal?

  • The prefrontal cortex is included in the sentence, as it indeed is important, also citation has been added

Line51: revised according to suggestions

Line54: revised according to suggestions 

Comment: It would be appropriate to indicate all the references that allowed the authors to make the table in Fig 1

  • This would be very hard to do, as table one was made based on studies of individual neuropeptides mentioned further in the article and there would be dozens of references. If it would be alright, we will rather stay, that Figure 1 was created based on all the data that follows in the article.

From line 57 to 63: long and disconnected chapter I would move it at the end and reformulate

  • We decided to remove most of this part so as not to distract from the main topic and the leftover was moved to chapter 2 neuropeptides

The whole introduction should be reformulated: I would start with the systematic data and then introduce the neuropeptides, that are the topic of the following chapter.

  • The introduction has been reordered according to suggestions and partially reformulated with some parts removed for clarity and less disconnect

comment: In table 1 are listed the neuropeptides with their precursor genes and receptors, but for some of them more receptors exist. Was it a choice of the authors to focus the attention only on the receptors involved in anxiety and depression and omit the ones which don't have an exact or important function? If so, it should be specified. eg: for this neuropeptide a number of receptors exist, but just these few are known to play a role.

  • Yes, it was deliberate so as not to distract the table with too many receptors as even now there are many. Perhaps adding a sentence about not including receptors that are not connected to anxiety or depression would be sufficient.

Reviewer 2 Report

The authors revised the whole manuscript and answered all the questions.

Author Response

Thank you very much.